# Genetic Diversity and Structure of Persian Walnut (*Juglans regia* L.) in Pakistan: Implications for Conservation

**DOI:** 10.3390/plants11131652

**Published:** 2022-06-22

**Authors:** Ephie A. Magige, Peng-Zhen Fan, Moses C. Wambulwa, Richard Milne, Zeng-Yuan Wu, Ya-Huang Luo, Raees Khan, Hong-Yu Wu, Hai-Ling Qi, Guang-Fu Zhu, Debabrata Maity, Ikramullah Khan, Lian-Ming Gao, Jie Liu

**Affiliations:** 1CAS Key Laboratory for Plant Diversity and Biogeography of East Asia, Kunming Institute of Botany, Chinese Academy of Sciences, Kunming 650201, China; ephieawinomagige@mail.kib.ac.cn (E.A.M.); mcwambulwa@gmail.com (M.C.W.); wuzengyuan@mail.kib.ac.cn (Z.-Y.W.); luoyahuang@mail.kib.ac.cn (Y.-H.L.); hongyu@mail.kib.ac.cn (H.-Y.W.); 2Germplasm of Bank of Wild Species, Kunming Institute of Botany, Chinese Academy of Sciences, Kunming 650201, China; fanpengzhen@mail.kib.ac.cn (P.-Z.F.); zhuguangfu@mail.kib.ac.cn (G.-F.Z.); 3University of Chinese Academy of Sciences, Beijing 100049, China; 4Department of Life Sciences, School of Science and Computing, South Eastern Kenya University, Kitui 170-90200, Kenya; 5Institute of Molecular Plant Sciences, School of Biological Sciences, University of Edinburgh, Edinburgh EH9 3JH, UK; r.milne@ed.ac.uk; 6School of Biological Sciences, The University of Adelaide, Adelaide, SA 5005, Australia; raeeskhanbotanist@gmail.com; 7School of Ecology and Environmental Science, Yunnan University, Kunming 650091, China; hailing@mail.kib.ac.cn; 8Department of Botany, University of Calcutta, Kolkata 700019, India; debabrata@gmail.com; 9Department of Botany, Abdul Wali Khan University Mardan, KP, Mardan 23200, Pakistan; ikramullah@awkum.edu.pk; 10Lijiang Forest Biodiversity National Observation and Research Station, Kunming Institute of Botany, Chinese Academy of Sciences, Lijiang 674100, China

**Keywords:** microsatellite, *Juglans regia*, genetic diversity, genetic structure, conservation, Pakistan

## Abstract

Persian (Common) walnut (*Juglans regia* L.) is a famous fruit tree species valued for its nutritious nuts and high-quality wood. Although walnut is widely distributed and plays an important role in the economy and culture of Pakistan, the genetic diversity and structure of its populations in the country remains poorly understood. Therefore, using 31 nuclear microsatellites, we assessed the genetic diversity and population structure of 12 walnut populations sampled across Pakistan. We also implemented the geostatistical IDW technique in ArcGIS to reveal “hotspots” of genetic diversity. Generally, the studied populations registered relatively low indices of genetic diversity (*N*_A_ = 3.839, *H*_O_ = 0.558, *U*_HE_ = 0.580), and eight populations had positive inbreeding coefficient (*F*_IS_) values. Low among-population differentiation was indicated by AMOVA, pairwise *F*_ST_ and *D*_C_. STRUCTURE, PCoA and neighbor joining (NJ) analysis revealed a general lack of clear clustering in the populations except that one population in Upper Dir was clearly genetically distinct from the rest. Furthermore, the Mantel test showed no correlation between the geographic and genetic distance (*r* = 0.14, *p* = 0.22), while barrier analysis suggested three statistically significant genetic barriers. Finally, the spatial interpolation results indicated that populations in Ziarat, Kashmir, Dir, Swat, Chitral, and upper Dir had high intrapopulation genetic diversity, suggesting the need to conserve populations in those areas. The results from this study will be important for future breeding improvement and conservation of walnuts in Pakistan.

## 1. Introduction

The common walnut (*Juglans regia* L.; Juglandaceae) is a monoecious, wind pollinated, fruit tree with animal-based seeds dispersed. It is self-compatible but achieves a high rate of cross-pollination [1,2], mostly favoring protandry but sometimes protogyny, and usually has a diploid genome karyotype of 2n = 32 [3,4,5,6], although a karyotype of 2n = 34 has been found in one study [7]. Economically, *J. regia* is valued for its nutritious and medicinal nuts as well as high-quality timber [8,9]. Its nuts have a rich biochemical profile, comprising fats (52–70%), proteins (14–24%), vitamins, and minerals [10]. These biochemical components have been linked to reduced risk of respiratory, cardiovascular, and cancer-associated complications [11]. The numerous dietary and therapeutic benefits associated with walnuts have contributed to the expansion of its local and global demand, with Pakistan producing approximately 0.4% of the global volume in 2021 [12]. Although Pakistan is an exporter of in-shell walnut, the production of the fruit is primarily for home consumption and local trade. 

It is thought that *J*. *regia* originated in the mountains of Eastern and Central Asia including Pakistan, even though the precise area of origin is still debatable [13,14,15]. Today *J*. *regia* is distributed across over 60 countries throughout the subtropical and temperate regions of the world, and is harvested from both cultivated and wild stands [16], thus it contributes greatly to the economy of those countries. However, despite the immense contribution of *J*. *regia* to the culture and economy of Pakistan, the country has been experiencing a declining trend in its production [17]. This problem can be addressed by scientific interventions such as germplasm improvement, marker assisted breeding, and *in situ* conservation. Specifically, a comprehensive analysis of the genetic diversity of walnut in Pakistan would provide a rational basis for parental selection of breeding materials, especially in the face of climate change and biotic stressors such as pests which threaten cultivated walnuts. This would ensure increased production to meet both the national and international market demand. Additionally, mapping out patterns of genetic diversity will help to determine areas that require urgent *in situ* and *ex situ* conservation of *J*. *regia*. Thus far, 249 individuals of *J. regia* from Pakistan have been investigated using a variety of methods, i.e., protein content (20 individuals [18]), morphology (203 individuals; [19]) fruit properties and nutritional composition (six cultivar individuals; [10]) and RAPD markers (20 individuals [19]). Morphological, physiological, and biochemical traits are all largely influenced by environmental factors and therefore unreliable indicators of genetic content or relatedness [20,21]. On the other hand, DNA-based molecular markers are stable, more reliable, and less prone to environmental plasticity, and are therefore preferred for population genetics studies. Neutral markers such as microsatellites (SSR) are ideal for genetic diversity studies due to their reproducibility, high polymorphism, codominant inheritance, and abundance in the genome [22]. Additionally, SSRs also exhibit transferability across species of *Juglans* and high universality across their genomes [23], as shown by the 32 novel SSR markers initially developed for *J*. *sigillata* [24], used successfully with other *Juglans* species [25]. 

SSR loci in *Juglans* have proved effective for the analysis of genetic diversity and population structure of both cultivated and wild walnut species [26,27,28,29,30]. However, only limited information exists on the genetic diversity and population structure of *J*. *regia* based on SSR markers in Pakistan [31]. Therefore, in the current study, we employed 31 SSR loci to genotype wild walnut trees in Pakistan. We aimed to (1) characterize the genetic diversity and structure of populations of *J*. *regia*, and (2) identify ‘hotspots’ of genetic diversity that deserve conservation. The results reported herein will provide a more detailed information on the genetic diversity and structure of walnut in Pakistan, and hence might promote effective and sustainable utilization and conservation of walnut germplasm in the country. 

## 2. Materials and Methods

### 2.1. Population Sampling

We collected 198 individuals of *J*. *regia* from 12 populations majorly distributed in natural forests along river sides and mountainous areas across Pakistan, even though some populations such as HCR were found in a settlement area (Table 1; Figure 1). These populations were collected between 2017 and 2019, and were treated as natural, based on the knowledge of the local people and the collectors. However, bearing the sampling sites of populations such as HCR, we clearly could not rule that sampled populations could be a mixture of wild and once cultivated walnut trees in unknown proportion. For each population, 9–22 individuals were collected, with a distance of at least 100 m between sampled individuals, and a population distance bearing a minimum of at least 1 km and maximum 700 km (Appendix A). Mature healthy leaves intended for DNA extraction were collected and dried in silica gel. Voucher specimens of each sampled individual were prepared and deposited at the herbarium of Kunming Institute of Botany, Chinese Academy of Sciences (KUN).

### 2.2. DNA Extraction, Microsatellite Amplification, and PCR Product Analysis

Total genomic DNA was extracted from about 0.02 g of silica gel-dried leaf tissue following a modified CTAB protocol [32,33], and concentrations of DNA were adjusted to 30–50 ng/µL for each sample. From a panel of screened primers, a total of 31 SSR primer pairs were selected (Appendix A), comprising 13 nuclear genomic SSRs originally developed for *J*. *sigillata* [24,25], four EST-SSRs developed for *J*. *regia* [34,35], one transcriptome-based SSR developed for *Juglans mandshurica* [36], two genomic library enrichment based SSRs designed for *J*. *regia* [37], and 11 Genome-based SSRs which have been used in our previous study [25]. These primer pairs included those with tri (20), tetra (7), penta (3), and hexa (1) repeat motifs (Appendix A), and all were used to genotype the 198 walnut trees. The forward primers were fluorescently labeled with FAM, HEX, or TAMRA dyes (Optimus Bio, Kunming, China) at the 5′end. Multiplexing based on color and size allowed us to group the 31 primer pairs into five multiplexes (Appendix A). PCR amplification was conducted on a Veriti^®^ 96-Well Thermo-Cycler (Applied Biosystems, Foster City, CA, USA). The 15 µL multiplex PCR mix comprised 1 µL of each reverse and forward primer, and 2 µL of the DNA template with the remaining volume topped up with an appropriate amount of Golden Star T6 Super PCR mix (Tsingke, Wuhan, China). The following thermocycling regimen was employed: initial denaturation at 98 °C for 2 min, 35 cycles 98 °C for 10 s, primer annealing temperatures (53–61 °C; Appendix A) for 15 s, 72 °C for 10 s, then a final extension at 72 °C for 5 min, with a holding temperature of 4 °C. The fragment sizes of PCR products were determined using an ABI 3730xl (automated sequencer Applied Biosystems, Foster City, CA, USA). GENEMARKER v4.0 (SoftGenetics, State College, PA, USA) was used to score the SSR data as diploid genotypes.

### 2.3. Data Analysis

MICROCHECKER v2.2.1 [38] was used to examine the presence of null alleles and allele dropout. We used GenAIEx v6.5 [39] to determine the percentage polymorphism per population (%P), attained by dividing the number of polymorphic bands in each population by the average number of bands. 

For each population, the number of alleles (*N*_A_), observed heterozygosity *H*_O_), expected heterozygosity (*H*_E_) and unbiased expected heterozygosity (*UH*_E_) across loci was calculated in GenAIEx. Any significant shift from Hardy-Weinberg equilibrium with respect to heterozygosity deficit was evaluated by testing for inbreeding coefficient (*F*_IS_) among the populations, with 5000 randomizations using FSTAT v2.9.3.2 [40]. Number of alleles or allelic richness of a given population can be used to gauge the population’s breeding prospects. However, the number of alleles per locus is often dependent on population size, hence allelic richness was calculated using rarefaction with HP-Rare v1.1 [41]. The private alleles numbers were calculated in GenAIEx. Subsequently, we used the Inverse Distance Weighted (IDW) interpolation function in the GIS software ArcGIS v10.7 (ESRI, Redlands, CA, USA) to infer the allelic richness and expected heterozygosity outside the sampled sites and areas. Additionally, we tested for correlation between genetic diversity (observed heterozygosity) and elevation using the R package *hierfstat* [42].

To examine the pairwise genetic differentiation (*F*_ST_) and genetic distance (*D*_C_) between pairs of populations, we used FreeNA v11.0 [43]. *F*_ST_ values were calculated based on the corrected ENA (excluding null alleles) procedure, and corrected values of Cavalli-Sforza and Edwards algorithm [44], while the calculation of the values of genetic distance (*D*_C_) followed the INA (including null alleles) algorithm. Subsequently, the data was represented on a heatmap generated in the Origin v8.0 program (Origin Lab Inc., Northampton, MA, USA). Analysis of molecular variance (AMOVA) was conducted in Arlequin v3.5.1.3 [45] to determine genetic variation among and within the populations. To infer the genetic structure of *J*. *regia* populations, the non-rooted tree was generated using the neighbor-joining (NJ) procedure with 1000 bootstrap replicates in POPTREE v2.0 [46]. Tree topologies were viewed and adjusted accordingly in Figtree v1.4.2 [47]. Bayesian analysis of population genetic structure with STRUCTURE v2.3.4 was used with the admixture model and a correlated allele frequency procedure [48,49]. To determine the optimal number of clusters (*K*), population structure was tested at *K* values ranging from 1 to 10, each with 10 replicates based on 100,000 Markov Chain Monte Carlo (MCMC) iterations following a burn-in period of 10,000 steps, followed by evaluation of optimal *K* using STRUCTURE HARVESTER v6.94 [50]. The population structure was displayed graphically using DISTRUCT v1.0 [51]). Principal coordinate analysis (PCoA) based on the covariance standardized method of pairwise Nei’s genetic distance implemented in GenAIEx was used to further determine the genetic structure of the studied walnut populations. Isolation by distance was determined by performing a Mantel test [52] in GenAIEx using a permutation of 1000. We used Geographical Distance Matrix Generator v1.2.3 [53] to generate the geographic distance matrix, whereas the *F*_ST_ matrix was obtained in GenAIEx. We also performed genetic barrier analysis using BARRIER v2.2 [54] following the maximum difference algorithm from [55], and with an assessment of five genetic barriers, to define the spatial differentiation between populations, and identify substantial breaks. The geographical coordinates and pairwise genetic distance were generated by MSA v4.05 [56] with a bootstrap of 100; these were used to link Delaunay triangulation and to infer the corresponding Voronoi tessellation. Nei’s genetic distance (*D*_A_); [57] was used to connect each edge of the Voronoi polygon. 

## 3. Results

### 3.1. Genetic Diversity

The SSR loci in our study showed high polymorphism, minimal stuttering, minimal allele drop-out, and low average number of null alleles (Appendix A). The mean percentage of polymorphic loci per population was 97%, with a range of 90% to 100%. The overall genetic diversity indices were generally low (*N*_A_ = 3.793, *H*_O_ = 0.563, *H*_E_ = 0.558). At population level, mean observed heterozygosity (*H*_O_) ranged from 0.523 (DIR) to 0.649 (ZTR) while the expected heterozygosity (*UH*_E_) was lowest in DUB (0.465) and highest in HDR (0.629). The average number of alleles for the entire dataset was 3.793, while the total number of alleles per population ranged from 83 (DUB) to 142 (HDR). Generally, the mean value for *UH*_E_ was slightly higher than the *H*_O_ (Table 2). All populations had private alleles, albeit in varying levels (data not shown). The mean inbreeding coefficient (*F*_IS_) ranged from −0.134 in DUB to 0.120 in HDR, with 7 of 12 populations showing positive inbreeding values (Table 2). 

Through geospatial interpolations of *UH*_E_, and *A*_R,_ we generated the genetic diversity landscape surfaces of *J*. *regia* in Pakistan (Appendix A). The highest values of *UH*_E_ (0.63) were reported in populations located in the southwestern and northern areas, whereas two populations in Dir 1 (DIR) and upper Dir (DUB), both in the northern region, registered the lowest *UH*_E_ values (0.47; Appendix A). *A*_R_ values ranged between 1.93 (DUB) and 2.39 (HDR). The highest values of A_R_ were mostly in the northern region, although two populations, one from the southwestern and the other from the Northeastern also registered high values of A_R_. As with the case of *UH*_E_, two populations from Dir and upper Dir also had low *A*_R_ levels in the range of 1.93–1.94 (Appendix A). The correlation analysis indicated a weakly positive but non-significant relationship between genetic diversity and elevation (Appendix A).

### 3.2. Population Genetic Structure 

#### 3.2.1. Patterns of Genetic Differentiation

Generally, the current populations had low (0.05 to 0.25) *F*_ST_ and moderate (0.20 to 0.50) *D*_C_ (Figure 2a). Population DUB was genetically divergent from the rest of populations (*F*_ST_ = 0.20–0.25; *D*_C_ = 0.42–0.52). On the other hand, certain population pairs (HDR-HCR, KAR-CLR and ZTR-CLR) appeared to be genetically close to one another in terms of both genetic differentiation and genetic distance. AMOVA analysis indicated that 9% of the total variation was partitioned among populations, 2% among individuals within populations, while the remaining 89% resided within individuals (Table 3).

#### 3.2.2. Two Main Genetic Groups

The neighbor-joining (NJ) tree partitioned the *J*. *regia* populations into two genetic clusters: cluster 1 comprising only one population (DUB) and cluster 2 comprising all other populations (Figure 2b). Populations HDR and HCR formed a sub-cluster within cluster 2 with high bootstrap support of >95%. STRUCTURE analysis showed that 2 was the best *K* value, followed by 3 (Figure 3b), hence both are shown (Figure 3a). Consistent with NJ results, under *K* = 2, population DUB was genetically distinguishable and separated from the rest of the populations in the STRUCTURE analysis (Figure 3a). PCoA likewise placed all members of DUB into one distinct cluster whereas all of the remaining 11 populations formed another loose cluster with some outliers (Figure 3c), further supporting the STRUCTURE and NJ results. STRUCTURE did, however, indicate that HCR and HDR were closer to DUB than were the other 9 populations. The clustering of populations in the NJ, STRUCTURE and PCoA analyses did not follow patterns of geographical distribution except that UKR and KMR were both geographically close and clustered together.

#### 3.2.3. Mantel Test and Genetic Barrier 

The Mantel test revealed a lack of significant correlation between geographic and genetic distance among the studied populations of *J*. *regia* (*r =* 0.14, *p* = 0.22) (Appendix A). However, three main genetic barriers were detected, with a bootstrap support of >79% (Figure 4). The first barrier (B1) separated the western populations UKR, KMR, DIR, DUB, HCR and CLR from populations SHR, HSR and KAR in the center of *J. regia*’s range in Pakistan. B2 mainly separated populations HDR from STR from the rest of the populations, while B3 separated ZTR from the rest. 

## 4. Discussion

### 4.1. Genetic Diversity of Walnut Populations

Our results show that populations of *J. regia* from Pakistan harbor a low to moderate level of genetic diversity (Table 2), which is generally comparable to the previous genetic diversity estimate for *J*. *regia* populations from other regions [29]. However, a recent study of cultivated *J*. *regia* from Western Himalaya, Pakistan revealed a higher heterozygosity value (0.227) [58], which was attributed to seed-based propagation, high heterozygosis and dichogamy. The relatively low genetic diversity in the current study could be due to habitat fragmentation, natural geographic barriers and/or anthropogenic habitat loss. Indeed, habitat fragmentation leads to population size reduction, consequently causing reduction in allelic richness and genetic diversity through genetic drift, increased selfing and mating among close relatives [59,60]. This phenomenon has been observed for other plant groups such as Yew plant [61] and *Spondias purpurea* [62].

Heterozygosity deficit were detected in eight *J*. *regia* populations. Three possible causes are the presence of null alleles, inbreeding, and population sub-structuring [63,64]. Null alleles result in underestimation of observed heterozygosity and overestimation of fixation indices [65]. Even though 14 loci had null alleles, their frequencies were moderately low (Appendix A), suggesting the role of other factors such as endogamy, and population sub-structuring. Inbreeding could be common in walnut populations, likely due to limited dispersal of the seeds, favoring mating with siblings or close relatives. Additionally, habitat fragmentation could also result in isolation of populations, causing population sub-structuring, ultimately impacting the number of heterozygotes. The observed heterozygote deficit further supports the observed low genetic diversity in the studied populations. Similar results were found in walnut populations from Xizang [66]. However, a further possible cause is artificial selection, if some populations include land races with the history of cultivation.

The interpolation of *UH*_E_ and *A*_R_ revealed high intrapopulation genetic diversity in the northern, eastern and southwestern regions suggesting that these geographical areas might be centers of genetic diversity for *J*. *regia* in Pakistan. The high levels of *UH*_E_ and *A*_R_ in the northern region indicate a possible center of genetic diversity, perhaps originating as a glacial refugium, as corroborated by the presence of private alleles in all the populations. Indeed, northern Pakistan might encompasses the Pleistocene refugial to where many plant species retreated during the ice ages; this area is therefore expected to harbor high genetic diversity [67,68]. However, without information on the fossils in Pleistocene of common walnut trees, this potential cause remains speculative for *J. regia*. Alternatively, considering the long-standing economic and cultural importance of walnut cultivation in Asia [69], the high genetic diversity of walnut in northern Pakistan could also be linked to admixture caused by artificial seed selection and dispersal. This also holds for populations KAR and ZTR in the southwestern and Northeastern regions of Pakistan, respectively. Additionally, the increase of genetic diversity with rising elevation is consistent with the view that higher elevations experience less human encroachment, hence have less ecosystem disturbance leading to better plant adaptability [70]. However, considering that this relationship was only weakly positive and non-significant, more comprehensive sampling may be required in order to arrive at a more definite conclusion in the future.

### 4.2. Genetic Structure of Walnut Populations

The AMOVA analysis revealed higher among-population than within-population partitioning genetic variation, a rare scenario particularly for species with outcrossing mating system such as walnut. Therefore, this could be as a result of habitat fragmentation as discussed above, and possibly the influence of artificial selection on landraces, if it occurs. Fragmentation of habitats limits or prevents long-distance pollination events, resulting in pollination only within clumps of close relatives. The results from AMOVA were consistent with the differentiation depicted by pairwise *F*_ST_ and *D*_C_ which suggested low to moderate *F*_ST_ and *D*_C_ within populations, similar to results obtained for *Calotropis gigantea* and *J*. *regia* from Kyrgyzstan [29,71].

STRUCTURE, PCoA and NJ tree results clearly demonstrated that population DUB was delineated from the rest of the populations, suggesting that gene flow from DUB to other populations was very limited, though HCR and HDR might have limited gene flow with DUB according to STRUCTURE. A normal explanation would be geographical barriers, but DUB is situated in the middle of a cluster of populations in northern Pakistan, whereas ZTR and KAR are far more distant. Therefore, the distinctness of DUB likely reflects its history, not its current geography. Hence DUB might have been genetically altered by gene flow from cultivated material or local land races, or possibly derive wholly from a distinct region (e.g., East Asia). Hence further research is needed on this population for better determination of its nature. The regional intermixing of populations other than DUB, as confirmed by STRUCTURE, PCoA and NJ tree appears inconsistent with a naturally fragmented distribution. The close proximity of many northern populations might explain the genetic similarity between them, but UKR and KMR are >300 km away from the main cluster, whereas KAR and ZTR are each >700 km away from all others. Hence the genetic similarity of these distant populations is difficult to explain in terms of the current distribution. Therefore, other factors must be considered, such as a common historical gene pool and previous connections in the range that have now been lost, or possible recently human or natural transport of propagules.

The Mantel test showed no correlation between geographical and genetic distance, thus further supporting our assumption of frequent gene flow and potential long-distance dispersal of propagules. Considering the topographical complexity of Pakistan, it is likely that walnut populations in the country are geographically isolated by the geographic barriers. Our analysis identified three significant genetic barriers, which might coincide with mountain ranges such as the Hindu Kush, Karakoram, and Himalaya in the northern Pakistan. Barrier B1 ran north to south, whereas B2 ran Northwest to Southeast. The third barrier, B3, might have been simple distance, or the varying relief (probably part of Chaghi hill) separating ZTR from the other populations (Figure 1). The presence of the geographical barriers among the studied populations suggests potential constraints to migration and natural gene flow, restricting outcrossing and promoting inbreeding and also selfing, because *J*. *regia* is self-compatible [72]. Thus, it is plausible that geographically imposed barriers helped shape the genetic structure of the studied populations, as has been shown for other taxa [73,74,75].

### 4.3. Genetic Diversity Hotspots and Conservation Implications 

The ultimate aim of conservation study is to guarantee the continuous existence of populations and to ensure the maintenance of their evolutionary potential [63]. Knowledge on the present levels of genetic diversity and the pattern of genetic variation of *J. regia* in Pakistan is vital to devise appropriate measures for conservation [76]. Genetic diversity is low for the populations of walnut in Pakistan, making them prone to extinction particularly if they occur in isolation. Therefore, we recommend both *in situ* and *ex situ* conservation strategies as a way of maintaining the adaptive potential of this species. We found that population DUB has a small population size, showed the least genetic diversity, and is also the population most distinct from others. Hence, it requires urgent attention through interventions by both *in situ* and *ex situ* conservation. Additionally, spatial interpolation of genetic diversity indices revealed the need to conserve populations HDR, HCR, KAR, SHR, ZTR and KAR on the basis of their high genetic diversity. Finally, we identified three significant barriers, which are constraints to migration and natural gene flow in the region. We recommend deliberate augmentation of geneflow within and between walnut populations through artificial dispersal strategies, except population DUB so as to avoid outbreeding depression.

## 5. Conclusions

The current study is the first attempt to comprehensively explore the genetic diversity and structure of *J*. *regia* in Pakistan. Our results suggested that the genetic diversity of *J*. *regia* in Pakistan is somewhat low. Additionally, our results revealed high genetic diversity in six populations, and genetic distinction in DUB therefore, conservation attention should be given to those populations to ensure ultimate germplasm improvement for future breeding. As a future research direction, we propose wider sampling of *J*. *regia* populations from adjacent regions, for better understanding of the genetic relationship of *J*. *regia* in Pakistan and adjacent regions. Moreover, we recommend seeking local knowledge regarding human intervention into apparently wild populations. Despite the limitations of our study, the data presented herein is a strong basis for the germplasm improvement and conservation management of *J*. *regia* in Pakistan. 

## Figures and Tables

**Figure 1 plants-11-01652-f001:**
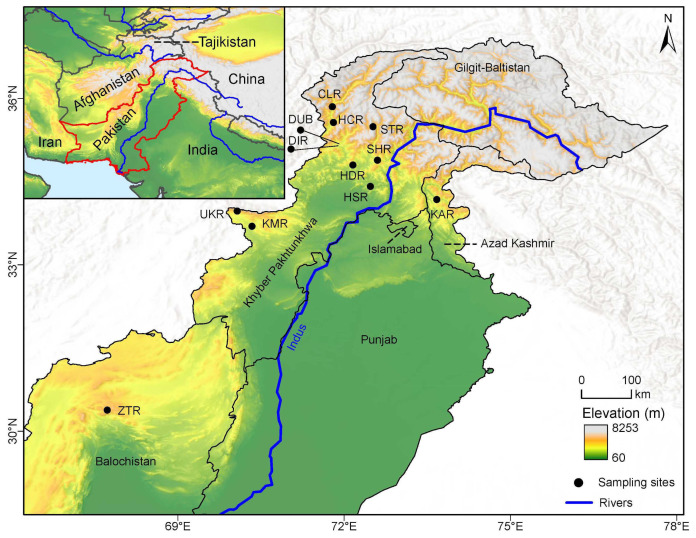
Sampling map of the 12 populations of walnut in this study. Sampling locations are indicated by black dots.

**Figure 2 plants-11-01652-f002:**
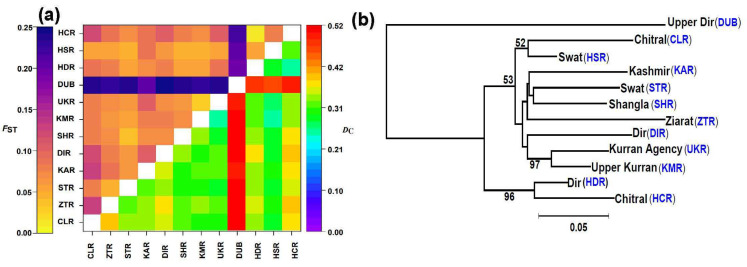
Genetic differentiation, distance and phylogenetic relationship among the studied populations. (**a**) Heat map for the pairwise genetic distance among populations. The upper part represents *F*_ST_ while the lower part denotes the *Dc* (Cavalli-Sforza). The values were computed based on 1000 permutations. (**b**) Neighbor-joining tree showing genetic relationships among 12 populations of walnut produced from 31 SSR markers.

**Figure 3 plants-11-01652-f003:**
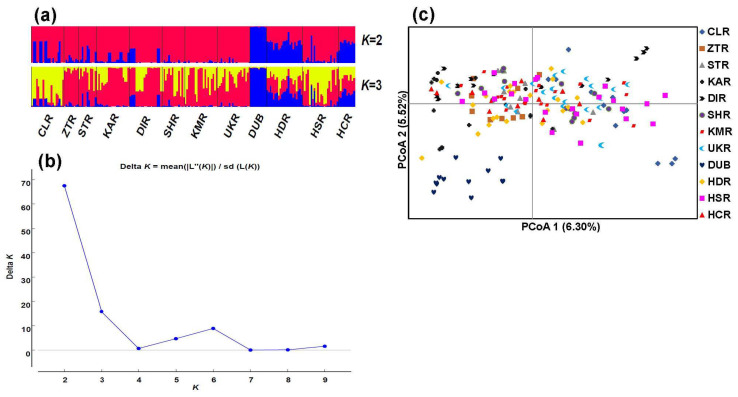
Genetic clustering of 12 populations of *J*. *regia* based on 31 SSR loci. (**a**) Bayesian inference of population structure at *K* = 2 and *K* = 3 using STRUCTURE. (**b**) Inference of the optimal *K* value using the Delta *K*. (**c**) Relationship among the walnut populations represented by the first two coordinates of the PCoA.

**Figure 4 plants-11-01652-f004:**
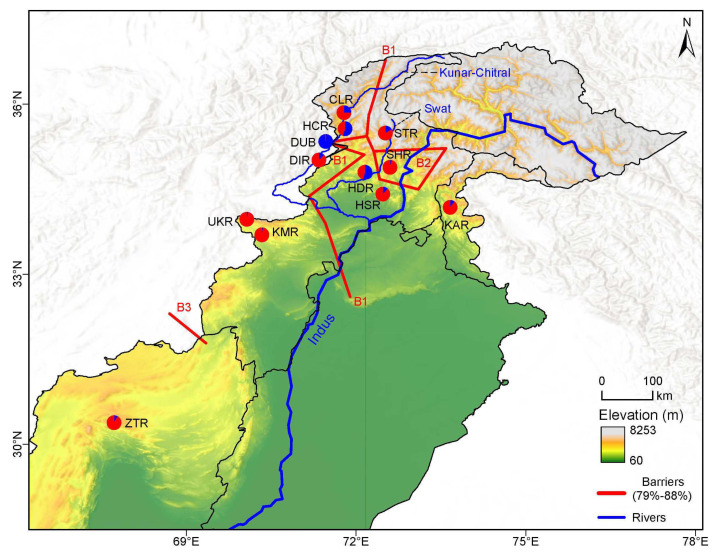
Spatial genetic structure of walnut in Pakistan. Pie color and proportion is in accordance to with genetic structure analysis results (Figure 3a). The three main detected barriers (B1, B2 and B3) are shown in red line, with the thin black lines indicating the administrative boundaries.

**Table 1 plants-11-01652-t001:** Sampling information of 12 populations of *J*. *regia* from Pakistan.

Sampling Site	ID	*N*	Latitude	Longitude	Elevation (m)	Habitat
Chitral	CLR	20	35.7713	71.7418	2753	Mountainous region
Ziarat	ZTR	9	30.3814	67.7179	2543	Mountainous region
Swat	STR	11	35.506	72.5714	2075	Mountainous region
Kashmir	KAR	20	34.1809	73.6642	1152	Mountainous region
Dir 1	DIR	20	35.1975	71.8681	1487	Mountainous region
Shangla	SHR	14	34.8873	72.6009	1904	Mountainous region
Kurram Agency	KMR	20	33.6959	70.3368	1197	River side
Upper Kurram	UKR	20	33.9702	70.0701	2964	Mountainous region
Dir upper	DUB	10	35.2119	71.8725	1514	Mountainous region
Dir 2	HDR	22	34.8012	72.1575	1245	Hilly areas
Swat	HSR	22	34.4147	72.4735	980	Hilly areas
Chitral	HCR	10	35.5688	71.8067	1359	Hilly areas

Note: ID, population identity, *N* number of collected individuals.

**Table 2 plants-11-01652-t002:** Statistical diversity parameters of walnut from Pakistan derived from 31 SSRs.

POP	*N* _T_	*N* _A_	*H* _O_	*H* _E_	*U* _HE_	*A* _R_	*F* _IS_
CLR	112	3.613	0.564	0.541	0.555	2.200	−0.016
ZTR	113	3.645	0.649	0.594	0.630	2.360	−0.033
STR	116	3.742	0.554	0.575	0.609	2.340	0.090
KAR	128	4.129	0.628	0.591	0.606	2.320	−0.038
DIR	99	3.194	0.509	0.521	0.535	2.110	0.049
SHR	122	3.935	0.610	0.580	0.602	2.320	−0.014
KMR	120	3.871	0.563	0.568	0.583	2.230	0.036
UKR	121	3.903	0.520	0.521	0.534	2.140	0.029
DUB	83	2.677	0.523	0.440	0.465	1.930	−0.134
HDR	142	4.581	0.542	0.614	0.629	2.390	0.141
HSR	133	4.290	0.514	0.569	0.582	2.260	0.120
HCR	122	3.935	0.575	0.578	0.609	2.340	0.060
Total	-	-	-	-	-	-	-
Mean	117.583	3.793	0.563	0.558	0.578	-	─

Note: Genetic diversity index: *N*_T_ total number of alleles, *N*_A_ number of alleles, *H*_O_ observed heterozygosity, *H*_E_ expected heterozygosity, *UH*_E_ unbiased expected heterozygosity, *A*_R_ allelic richness, *F*_IS_ inbreeding coefficient.

**Table 3 plants-11-01652-t003:** Hierarchical AMOVA results among 12 walnut populations generated from 31 SSR loci.

Source of Variation	Degree of Freedom	Sum of Squares	Variance Components	Percentage Variation (%)	*p* Value
Among populations	11	383.111	0.588	9	<0.001
Among individuals	186	1577.53	0.106	2	<0.001
within populations					
Within individuals	198	1593.5	6.280	89	<0.001
Total	395	3554.14	6.974	100	

## Data Availability

The SSR data generated in the current study have been provided in Appendix A.

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
