# Peer review of "Genetic Diversity and Structure of Persian Walnut (Juglans regia L.) in Pakistan: Implications for Conservation"

_plants, 2022, doi:10.3390/plants11131652_

Round 1
Reviewer 1 Report
Dear Author
Thank you for submitting your valuable manuscript to our journal. I understand you carried out genotyping walnut germplasm collected in Pakistan and clarified the genetic diversity and structure. But I also think there are many points which should be added in your manuscript. Please refer to the following points and modify your manuscript.
Major points
You used only 31 SSR markers to evaluate genetic structure of the materials. I think the selection of DNA markers is quite critical in the phylogenetic analysis, because the conclusion would be turn different if the markers are not suitable for evaluating genetic conformation of the materials. In my opinion, only 31 markers are insufficient to evaluate extensive and detailed genetic conformation of the materials (only simple evaluation-cluster analysis and PCoA analysis- are acceptable.).
Recently Chandler v2.0 (New version of reference genome of Persian walnut) was developed by Marrano et al. (2019), then I think you can indicate the physical location of the SSR markers you used referring to their study. In addition to this, please mention further information to confirm the suitability of these 31 markers for evaluating genetic structure.
With respect to the SSR markers, only size range was noted in Table S1. In order to confirm the reproducibility of this study, you should post the ‘Number of alleles’ and ‘All the allele types (e.g. 128, 130, 135, null)’ in this table. In addition to this information, you have to prepare another supplemental table, which shows allele information of the used materials. Please refer to the following table.
Sampling site |
Plant ID |
Allele pattern of SSR marker (bp) |
|
JUGNEW21 |
J11 |
||
CLR |
1 |
128/130 |
235/268 |
|
2 |
130/130 |
235/240 |
ZTR |
1 |
130/135 |
268/268 |
|
2 |
null/null |
250/255 |
|
3 |
155/155 |
235/240 |
According to Table 4, genetic variation among individuals is quite larger compared with the variation among populations. I am wondering what kind of criteria you used to classify these 12 populations before starting this study (Some sampling sites are very close to each other. What is the difference of these neighboring sites?). Please elaborate.
With regards to the above-mentioned point, you only mentioned genetic diversity and structure information in this manuscript. But I imagine morphological and physiological variations would contribute to the classification of 12 populations in addition to the geographical distance. Please provide those kinds of information including figures of nuts (if there are differences of shapes and colors among populations).
You mentioned null allele was detected in four markers in discussion section. I think this point is quite important because it is impossible to discriminate between homozygote alleles and heterozygote alleles in case of SSR markers (e.g. 238/238 alleles and 238/- alleles are impossible to distinguish in fragment analysis). I imagine there is a possibility that null allele could exist in other SSR markers. Please elaborate this point in discussion section.
Minor point
There is a same name of location, ‘DIR’ in Table 1 (but abbreviation is different from each other, DIR and HDR). This expression might lead to the misunderstanding of the content of this manuscript. You should change the either of the name.
The name of ‘DUR’ in Figure 1 should be changed to ‘DUB’ according to Table 1 and Figure 5.
Reviewer 2 Report
The study describes patterns of genetic diversity and differentiation of walnut populations in Pakistan. The work is entirely based on an analysis of variation at SSRs.
Sampling:
Sample sizes range from 9 to 20 trees per population, which is very low for an investigation based on highly polymorphic markers at SSRs. Reasons should be given for the uneven sample sizes in the different populations. No details concerning potential human impact, approximate population size, human impact, natural regeneration, tree height, tree health, fruiting or flowering and other adaptive traits at the level of populations or individuals are reported, severely limiting data interpretation in particular with regard to conservation. The geographic coordinates (latitude, longitude) correspond to the sampling sites as described in the first column of Table 1. However, a check of the location according to the coordinates as described in Table 1 with google maps reveals that the occurrence of natural walnut populations is in most cases not plausible. For example, population KAR is located in an urban area few kilometers from Lahore. Several other locations are also in cities or villages (e.g. CLR, ZTR, DIR, DUB and others). Population STR is located above 3000 m elevation according to its coordinates and not on a river side. Sampling needs to be described accurate and in more detail.
Methods and Results:
Multiplexing was used to genotype the plants. According to Table S1, five to seven different annealing temperatures were used for the three different fluorescent dyes. It is not clear how analyses were conducted with only five multiplexes since more annealing temperatures are described for some dyes. Furthermore, annealing temperatures vary from 53 to 61 degree C (and not from 50.6 to 61 degree C as written in the text). Table S1 should provide for each SSR locus information on the original publication, where the locus was described first, and on the pooling of the different loci for multiplexing.
It is not clear how NA was calculated (Tab. 2). Since 31 loci were analyzed, numbers for all alleles should be much higher than 3-5.
Figure 2 is not informative in view of the spatial distribution of populations.
Discussion:
The discussion is to a large extent very speculative due to the lack of information about the sampled populations apart from their variation at SSRs.
It is surprising that the authors refrain from comparing their results with the study of Torokeldiev et al. (2019): Torokeldiev, N., Ziehe, M., Gailing, O. et al. Genetic diversity and structure of natural Juglans regia L. populations in the southern Kyrgyz Republic revealed by nuclear SSR and EST-SSR markers. Tree Genetics & Genomes 15, 5 (2019). https://doi.org/10.1007/s11295-018-1311-8
Reviewer 3 Report
Dear Authors
The article contains data that is important for breeding and conservation of walnuts in Pakistan. The article in general is well written and the information presented in a logical manner. Therefore, the present draft needs minor revision before further process.
Detailed comments below
Literature must be changed to numbers
Keywords: semicolons instead of dots
M&M line 131, The 15μL multiplex PCR composed 0.2mL??? of each primer, must be explained
Results line 241 Figure 2, no mark b
Table 3, Abbreviations in column titles must be expanded

Reviewer 4 Report
In this manuscript, the authors propose a genotyping of twelve wild walnut populations of Pakistan by 31 SSR loci to characterize their genetic structure and diversity and to identify potential hotspots.
The authors approach this study by analyzing both inter-and intra-population diversity. A low genetic diversity was observed among the populations suggesting and recommending in-situ and ex-situ conservation strategies to maintain the adaptive potential of this species. The anthropogenic influence is not clear in some of these walnut populations, as is also reported in the literature for other Central Asian areas.
The manuscript has already undergone two revisions, and I could observe an improvement in the writing and content in this latest version. Nevertheless, I would recommend that the authors revise the Abstract, based on the two previous revisions of the manuscript, especially for:
Line 33-36:
All populations were bottlenecked according to the infinite allele model (IAM), with two populations from Dir and one from Upper Kurram further showing evidence of bottleneck under both stepwise mutation model (SMM) and two-phase mutation model (TPM).
Below are other specific points for the authors to consider and address:
Introduction
Line 49: ….fruit tree with animal-based seed dispersed.
Correct “seeds”
Line 53 and 54 :… well as high-quality timber [8, 9]). Its nuts have a rich biochemical profile, 53 comprising fats (52-70%), proteins (14-24%), vitamins, and minerals ([10]
delete the round bracket next to the square bracket
Line 69: “in-situ” and “ex-situ” should be written in Italic font. Please check the manuscript and correct it.
Line 71: …..especially in the face of climate change and biotic strressors such as pests which ……
Correct
Stressors
Line 80: …. environmenal factors…
Correct environmental
….content or relateness 80 [22, 23].
Correct
relatedness
Line 87-88: …. which have also been used successfully with other Juglans species [27].
used successfully with other Juglans species [27].
It is more fluent
Materials and methods
Line 96: 2.1. Population sampling
No bold font
Line 105:…. and were treated as natural. based on the….
replace dot with a comma
Line 136: ….of each reverse and forwardprimer, and…
Correct
forward primer
Line 139: …..regimen was employed: Initial denaturation at 98°C for 2min, 35 cycles 98°C for 10s……
Correct
regimen was employed: initial denaturation at 98°C for 2min, 35 cycles 98°C for 10s……
Results
Line 217-219: The highest values of AR were mostly in the northern region, although two populations, one from southwestern and the other from Northeastern also registered high values of AR.
Correct
The highest values of AR were mostly in the northern region, although two populations, one from the southwestern and the other from the Northeastern also registered high values of AR.
Discussion
Line 301: … ultimately impacting on the number of heterozygotes
Correct
ultimately impacting the number of heterozygotes.
Line 371: … vital step towards devising appropriate measures for conservation [77].
Correct
toward
Line 372: .. making them are prone to …
Correct
making them prone to …
I suggest to the authors follow the format of the Plants journal, following the Front Matter and Back Matter, with particular regard to the references, in the section "Instructions for Authors”
Reviewer 5 Report
1. In order to improve the paper and increase its practical value, I think it is necessary to mention some strong characteristics for each of the 12 walnut populations studied, such as: fruit size and quality, tree height, resistance to various factors biotic and abiotic stress etc. Table 1 could be one of the right places for this information.
2. Would it be interesting for the authors to mention the situation of the genetic diversity of cultivated walnuts and cultivated areas with this species in Pakistan.
Round 2
Reviewer 1 Report
Dear Author
I admit that your manuscript have been modified based on most of my suggestions. I hope you will further study the relationship between the genome structure of walnut cultivars and their practical agriculture traits in your further approach.
Author Response
Response to reviewer 1:
We thank the reviewer for taking time to go through our previously made corrections and suggestions and for offering to provide another round of feedback on our manuscript for its valuable improvement. We have incorporated the suggestions raised. Below is a point-by-point response to the suggestions and concerns.
Extensive editing of English language and style required
Authors’ response: We truly appreciate the reviewer for raising this suggestion. Indeed, a well written manuscript should be devoid of any grammatical error. However, we must admit that our manuscript was revised and has again been revised by a native English speaker. We believe it should now be okay for publication.
I admit that your manuscript has been modified based on most of my suggestions. I hope you will further study the relationship between the genome structure of walnut cultivars and their practical agriculture traits in your further approach.
Authors’ response: We thank you for revising our manuscript keenly, and appreciating our rebuttals. You have raised quite an important point which we will have to look into in our future walnut publications.
Reviewer 2 Report
Carefully conducted field work and a clear description of this part of the work is an important aspect of the type of genetic inventories reported here for walnut in Pakistan. Thus, it is good that the authors checked the geographic locations of their sample sites and identified numerous errors in the first version of the manuscript. Still, the geographic location of the sampling sites make it highly unlikely to sample natural walnut populations, for example at site Chitral (HCR), which is a densely populated area.
The authors admit that the field work had severe weaknesses, resulting in a mainly speculative discussion. Thus, the publication of the genetic study should wait until future studies of walnuts from Pakistan based on properly conducted field work have been conducted as planned by the authors.
Author Response
Thank you for giving us another opportunity to submit the revised version of the manuscript. We do not take for granted the time taken off the busy schedule for revising and providing the valuable feedback. We have tried incorporating the changes to improve our manuscript. Please, below is the response to the suggestions and concerns raised.
Carefully conducted field work and a clear description of this part of the work is an important aspect of the type of genetic inventories reported here for walnut in Pakistan. Thus, it is good that the authors checked the geographic locations of their sample sites and identified numerous errors in the first version of the manuscript. Still, the geographic location of the sampling sites makes it highly unlikely to sample natural walnut populations, for example at site Chitral (HCR), which is a densely populated area.
The authors admit that the field work had severe weaknesses, resulting in a mainly speculative discussion. Thus, the publication of the genetic study should wait until future studies of walnuts from Pakistan based on properly conducted field work have been conducted as planned by the authors.
Authors’ response:
Thank you for raising this concern once again. We also appreciate that you recognized our initial revision, particularly on the GPS locations of these populations. Having rechecked sampling sites of these populations, particularly for population HCR, the raised concern makes a lot of sense, and we absolutely agree with you. Despite the re-confirmation from the collector and local people about the nature of these populations, it was not easy to arrive at a more definite conclusion on the true nature of these populations, particularly because none of them were being tended to despite occurring in densely populated area. Therefore, we agree that the sampled populations could be mixtures wild and cultivated walnuts, though in unknown proportion. Besides, knowing the true wild germplasm has remained challenging to researchers, particularly in the cases of less record of the histories regarding the population. Also, we cannot rule out the existence of natural populations in the settlements area because sometimes people tend to occupy formally unoccupied. Moreover, as was pointed out in the previous revision, we believe that our study forms a good basis for any future walnut studies in Pakistan, just as other previously conducted studies on walnut from other regions (Pollegioni et al., 2014; Vahdati et al., 2015; Ebrahimi et al., 2011; Karimi et al., 2010). Nonetheless, we have revised the methodology part (on population sampling) to factor in some the suggestion raised.